# The Systemic Inflammation Index on Admission Predicts In-Hospital Mortality in COVID-19 Patients

**DOI:** 10.3390/molecules25235725

**Published:** 2020-12-04

**Authors:** Alessandro G. Fois, Panagiotis Paliogiannis, Valentina Scano, Stefania Cau, Sergio Babudieri, Roberto Perra, Giulia Ruzzittu, Elisabetta Zinellu, Pietro Pirina, Ciriaco Carru, Luigi B. Arru, Alessandro Fancellu, Michele Mondoni, Arduino A. Mangoni, Angelo Zinellu

**Affiliations:** 1Department of Medical, Surgical and Experimental Sciences, University of Sassari, 07100 Sassari, Italy; agfois@uniss.it (A.G.F.); valescano93@gmail.com (V.S.); stefaniacau.s@gmail.com (S.C.); babuder@uniss.it (S.B.); pirina@uniss.it (P.P.); afancel@uniss.it (A.F.); 2Unit of Respiratory Diseases, University Hospital Sassari (AOU), 07100 Sassari, Italy; elisabetta.zinellu@aousassari.it; 3Pneumology Unit, Santissima Trinità Hospital, 09121 Cagliari, Italy; r.perra@gmail.com; 4Mater Olbia Hospital, 07026 Olbia, Italy; giuliaruzittu@hotmail.com; 5Department of Biomedical Sciences, University of Sassari, 07100 Sassari, Italy; carru@uniss.it (C.C.); azinellu@uniss.it (A.Z.); 6Operative Unit of Hematology, Center for Stem Cell Transplantation, San Francesco Hospital, 08100 Nuoro, Italy; luiarru@gmail.com; 7Respiratory Unit, ASST Santi Paolo e Carlo, San Paolo Hospital, Department of Health Sciences, University of Milan, 20122 Milan, Italy; michele.mondoni@asst-santipaolocarlo.it; 8Department of Clinical Pharmacology, College of Medicine and Public Health, Flinders University and Flinders Medical Centre, Adelaide 5042, Australia; arduino.mangoni@flinders.edu.au

**Keywords:** CBC, coronavirus, COVID-19, inflammation, SII

## Abstract

Background. The rapid onset of a systemic pro-inflammatory state followed by acute respiratory distress syndrome is the leading cause of mortality in patients with COVID-19. We performed a retrospective observational study to explore the capacity of different complete blood cell count (CBC)-derived inflammation indexes to predict in-hospital mortality in this group. Methods. The neutrophil to lymphocyte ratio (NLR), derived NLR (dNLR), platelet to lymphocyte ratio (PLR), mean platelet volume to platelet ratio (MPR), neutrophil to lymphocyte × platelet ratio (NLPR), monocyte to lymphocyte ratio (MLR), systemic inflammation response index (SIRI), systemic inflammation index (SII), and the aggregate index of systemic inflammation (AISI) were calculated on hospital admission in 119 patients with laboratory confirmed COVID-19. Results. Non-survivors had significantly higher AISI, dNLR, NLPR, NLR, SII, and SIRI values when compared to survivors. Similarly, Kaplan–Meier survival curves showed significantly lower survival in patients with higher AISI, dNLR, MLR, NLPR, NLR, SII, and SIRI. However, after adjusting for confounders, only the SII remained significantly associated with survival (HR = 1.0001; 95% CI, 1.0000–1.0001, *p* = 0.029) in multivariate Cox regression analysis. Conclusions. The SII on admission independently predicts in-hospital mortality in COVID-19 patients and may assist with early risk stratification in this group.

## 1. Introduction

In December 2019, an ongoing outbreak of unexplained pneumonia in China gained global attention [1]. Gene sequencing allowed the identification of a novel β-coronavirus as the responsible agent, which the International Committee on Taxonomy of Viruses identified as severe acute respiratory syndrome coronavirus 2 (SARS-CoV-2). On 30 January 2020, the associated pneumonia was referred to as coronavirus disease 2019 (COVID-19) by the World Health Organization [2]. As of 28 October 2020, more than forty-four million confirmed cases and over 1,175,000 deaths were reported worldwide [3]. Severity of the disease in patients is generally categorized as mild, moderate, severe, and critical based on clinical symptoms and laboratory test results [4,5]. In the majority of cases (about 81%), patients experience mild/moderate symptoms such as low-grade fever and cough [6]. Patients with severe symptoms and critical cases comprise only 14% and 5%, respectively, of all infected subjects, but they can develop severe pneumonia, acute respiratory distress syndrome (ARDS), and multiple organ failure that requires hospitalization and may lead to death [7]. Indeed, almost 20% of hospitalized patients need transfer to the intensive care unit (ICU), where the mortality rate is 61.5% [8,9]. Therefore, the identification of early biomarkers of disease severity might facilitate early aggressive treatment, reducing mortality and improving hospital resource allocation.

Routine blood tests for the assessment of inflammatory processes are often useful in the early diagnosis of several diseases [10]. In particular, the complete blood count is easy to perform, inexpensive, and provides information regarding various cell types and morphological parameters, i.e., white blood count, lymphocytes, neutrophils, monocytes, platelet count (PLT), and mean platelet volume. In addition, combined ratios of these parameters are also used as inflammation indexes and have been proposed as biomarkers to assist with the diagnosis, progress, and risk stratification of inflammatory diseases [11,12,13,14,15]. Recently, the neutrophil to lymphocyte ratio (NLR), derived NLR (dNLR), platelet to lymphocyte ratio (PLR), monocyte to lymphocyte ratio (MLR), and systemic inflammation response index (SII) have been shown to be useful for the diagnosis and severity assessment of COVID-19 patients [10,16,17]. However, few studies have assessed the prognostic capacity of these indexes, and they were limited to the NLR, dNLR, and PLR [17,18]. We sought to address this issue by comparing the capacity of the NLR, dNRL, PLR, MLR, SII, NLPR, MPR, systemic inflammation response index (SIRI), and aggregate index of systemic inflammation (AISI) to predict mortality in COVID-19 patients admitted to hospital.

## 2. Results

A total of 119 COVID-19 patients (77 men and 42 women) were included in the study (Table 1). The median age was 72 (IQR: 60–81) years. Ninety patients (75.6%) were discharged alive, whereas the remaining 29 (24.4%) died. Of the 119 patients, 81 (68.1%) had one or more pre-existing diseases, of which cardiovascular disease, respiratory disease, and diabetes were the most frequent (between 58 and 21%). The median hospitalization duration was 15 (IQR: 10–26) days. As reported in Table 1, non-survivors were significantly older (80 years, IQR 74–85 years vs. 68 years, IQR 57–78 years; *p* < 0.001), suffered more frequently from cardiovascular disease (79% vs. 51%, *p* = 0.008), and had higher values of the Charlson comorbidity index (median: 6.0, IQR 5.0–8.0 vs. 3.5, IQR 2.0–6.0, *p* < 0.001). Survivors had significantly higher P/F ratio values (278 ± 90 vs. 208 ± 88), longer time interval between disease onset and admission (median: 7.0 days, IQR 4.0–9.0 days vs. 4.0 days, IQR 1.5–6.0 days, *p* = 0.006), and longer hospital stay (median: 20 days, IQR: 13–34 days vs. 6 days, IQR 4–12 days). A significant difference between survivors and non-survivors was also observed in intensity of care (*p* = 0.03). In addition, non-survivors had significantly higher values of white blood cells (WBCs) (median: 9.16 × 10^9^ L; IQR: 5.65–13.52 × 10^9^ L vs. 6.47 × 10^9^ L IQR: 5.00–8.66 × 10^9^ L, *p* = 0.006), neutrophils (median: 7.30 × 10^9^ L; IQR: 3.75–11.15 × 10^9^ L vs. 5.00 × 10^9^ L IQR: 3.56–6.30 × 10^9^ L, *p* = 0.012), and lower values of lymphocytes (median: 0.70 × 10^9^ L; IQR: 0.60–1.03 × 10^9^ L vs. 1.00 × 10^9^ L IQR: 0.70–1.20 × 10^9^ L, *p* = 0.012). By contrast, there were no significant differences between survivors and non-survivors in gender, body mass index (BMI), smoking status, chest CT severity score, kidney disease, respiratory disease, diabetes, autoimmunity disease, malignancies, use of angiotensin converting enzyme (ACE) inhibitors or angiotensin II receptor blockers (ARBs), monocytes, platelets, red cell distribution width (RDW), and mean platelet volume (MPV).

Combined blood cell count indexes of inflammation are reported in Table 2. Non-survivors had higher values of AISI (median; 952; IQR: 201–1943 vs. 382; IQR 3139–894, *p* = 0.025), dNLR (median; 6.33; IQR: 2.87–10.19 vs. 3.06; IQR 2.16–5.07, *p* = 0.008), NLPR (median; 0.0215; IQR: 0.0142–0.0508 vs. 0.0121; IQR 0.0075–0.0194, *p* = 0.0009), NLR (median; 9.17; IQR: 5.35–18.55 vs. 5.00; IQR 3.27–8.44, *p* = 0.0015), SII (median; 1854; IQR: 706–4566 vs. 1022; IQR 616–1896, *p* = 0.039), and SIRI (median; 3.11; IQR: 1.19–5.56 vs. 1.64; IQR 1.07–2.94, *p* = 0.010). By contrast, there were no significant between-group differences in MLR, MPR, and PLR.

With respect to survival, the optimal cutoff values identified by ROC analysis were as follows: AISI, 798; dNLR, 6.2; MLR, 0.364; MPR, 0.040; NLPR, 0.019; NLR15.2; PLR, 240; SII, 1835; SIRI, 2.93 (Table 3). The values of area under the curve (AUC) were 0.640 (0.546 to 0.726) for AISI, 0.664 (0.571 to 0.748) for dNLR, 0.617 (0.523 to 0.705) for MLR, 0.558 (0.464 to 0.649) for MPR, 0.706 (0.615 to 0.787) for NLPR, 0.697 (0.605 to 0.778) for NLR, 0.572 (0.478 to 0.663) for PLR, 0.628 (0.534 to 0.715) for SII, and 0.659 (0.567 to 0.744) for SIRI. The Kaplan–Meier survival curves, after classifying the patients on the basis of Youden cut-offs obtained by ROC curves (Figure 1), showed significant lower survival with higher values of AISI (HR = 3.53; 95% CI 1.61–7.78, *p* = 0.002), dNLR (HR = 7.33; 95% CI 2.94–8.30, *p* < 0.001), MLR (HR = 2.84; 95% CI 1.35–5.95, *p* = 0.006), NLPR (HR = 4.21; 95% CI 1.94–9.13, *p* < 0.001), NLR (HR = 108; 95% CI 26–450, *p* < 0.001), SII (HR = 3.29; 95% CI 1.48–7.32, *p* = 0.034), and SIRI (HR = 3.63; 95% CI 1.64–8.00, *p* < 0.001). The multivariate Cox regression models reported in Table 4 showed that only the SII was significantly associated with survival (HR = 1.0001; 95% CI, 1.0000–1.0001, *p* = 0.029) after correction for age, P/F ratio, intensity of care, and Charlson comorbidity index. A trend towards statistical significance was also observed with the dNLR (HR = 1.0926; 95% CI, 0.9943–1.2006, *p* = 0.066) and the PLR (HR = 1.0006; 95% CI, 1.0000–1.0013, *p* = 0.058).

Table 5 shows demographic, clinical, and hematological characteristics of COVID-19 patients stratified according to SII values. Patients with SII > 1835 had lower P/F ratios when compared with patients with SII ≤ 1835 (226 ± 79 vs. 301 ± 85, *p* < 0.0001); higher chest CT severity score (median: 21.0, IQR 13.3–23.0 vs. 9.0, IQR 5.0–14.0, *p* < 0.001); increased values of WBC (median: 9.90 × 10^9^ L; IQR: 7.49–12.60 × 10^9^L vs. 5.90 × 10^9^ L IQR: 4.8–7.87 × 10^9^ L, *p* < 0.001), neutrophils (median: 8.65 × 10^9^ L; IQR: 6.00–11.60 × 10^9^ L vs. 4.40 × 10^9^ L IQR: 3.40–5.55 × 10^9^ L, *p* < 0.001), and platelets (267 × 10^9^ L; IQR: 220–361 × 10^9^ L vs. 183 × 10^9^ L IQR: 147–233 × 10^9^ L, *p* < 0.001); and reduced values of lymphocytes (median: 80.64 × 10^9^ L; IQR: 0.50–0.80 × 10^9^ L vs. 1.00 × 10^9^ L IQR: 0.80–1.35 × 10^9^ L, *p* < 0.001). Furthermore, patients with higher SII values had increased mortality rate (42% vs. 16%, *p* = 0.002). No significant differences were observed in other variables.

## 3. Discussion

This study described 119 COVID-19 patients hospitalized at four referral centers in Sardinia (Italy) from 15 March to 15 May 2020, with clinical and demographic characteristics similar to those recently described in other COVID-19 cohorts [1,6,17,18,19,20,21,22,23,24,25,26,27,28,29,30,31,32,33,34,35]. The lag time between the onset of symptoms and admission to hospital, a crucial factor in the community transmission rate, was 6.5 days (IQR 3–8 days), within the range of previous reports, between 4 and 12 days [17,19,20,21,22,24,25,28,30,31]. The median length of hospital stay in our cohort (15 days, IQR 10–26 days) was also similar to that of recent reports (between 12 and 22 days) [25,29,31,32]. As previously reported, we found that disease severity was significantly related to age [6,18,25,27,28,29,32,33], cardiovascular disease [6,18,29,33,34], interval between disease onset and admission [29,32], and P/F ratio [35].

Several laboratory abnormalities were reported in COVID-19 patients, particularly in severe and critically ill patients [19,20,21,22,23,24,25,26,27,28,29,30,31,32,33,34,35,36,37,38,39,40,41]. In line with other studies, among hematological parameters, leukocytosis, lymphopenia, and increased neutrophil count were related to disease severity in our cohort [1,6,18,19,25,27,28,29,36,37,38,39,40,41]. Neutrophils, the most abundant circulating white blood cells, are an important component of the immune system. They represent the first line of the innate immune defense, since they play a fundamental protective role during infection from bacteria and fungi, by killing these microorganisms by phagocytosis as well as neutrophil extracellular trap (NET) formation. However, their role in viral infections remains unclear. In mice infected by SARS-CoV, neutrophils seem not necessary for virus clearance from pulmonary cells and host survival [42]. Lungs from autopsy of patients affected by COVID-19 showed extensive neutrophil infiltration in pulmonary capillaries with extravasation into the alveolar space. The presence of both acute capillaritis as well as trachea neutrophilic mucositis demonstrates widespread inflammation across the airways [42].

In addition, inflammatory cell accumulation associated with endothelial cells infection during COVID-19 disease may induce endothelitis in different organs, thus contributing to systemic damage of microcirculatory function and leading to the phenomenon known as “happy hypoxia” [42]. On the other hand, the lymphopenia observed in COVID-19 seems to be linked to the ability of the virus to infect T cells through the angiotensin-converting enzyme 2 (ACE2) receptors and CD147-spike protein [43]. Therefore, a decreased level of CD3+, CD4+, and CD8+ T lymphocytes and an increased number of regulatory T cells are frequently observed in COVID-19 disease.

The increase of proinflammatory cytokines during T cell lymphopenia predisposes severe COVID-19 patients to a cytokine storm, thus resulting in multi-organ failure and death. In general, T lymphocytes CD4+ and CD8+ decrease is associated with disease severity and leads to increased NLR values, which have been reported to be a more sensitive biomarker of inflammation than the individual levels of neutrophils and lymphocytes [43]. In agreement with previous reports, we found increased values of NLR and dNLR in severe COVID-19 disease patients [18,19]. However, we also report for the first time that other blood cell count-derived inflammation indexes such as NLPR, SII, SIRI, and AISI are also significantly associated with disease severity. In particular, in our study, the AUC values of both NLR and NLPR were the highest, among the combined indexes evaluated, in predicting disease severity (about 0.70). The data of NLR AUC agreed well with those of previous reports, ranging between 0.65 and 0.73 [44,45,46]. In addition, we found that the AUC was significant with the dNLR, AISI, and SIRI, and had borderline significant with the SII and MLR (*p* = 0.060 and *p* = 0.054, respectively).

Kaplan–Meier survival curves using cut-off values obtained from ROC curves showed that survival was significantly associated with AISI, dNLR, MLR, NLPR, NLR, SII, and SIRI (Figure 1). However, after correction for age, P/F ratio, intensity of care, and Charlson comorbidity index, confounders identified in univariate analysis, only the SII remained associated with survival after multivariate Cox regression analysis. The SII includes three peripheral blood parameters, namely neutrophil, platelet, and lymphocyte count, which comprehensively summarizes the balance of host immune and inflammatory status. It has been already suggested as a prognostic biomarker in sepsis patients [47]. In addition, the SII has also been shown to be associated with worse survival in small cell lung cancer, hepatocellular carcinoma, colorectal cancer, and gastric cancer [48,49,50,51]. Recently, it has also been reported that the SII was significantly altered in COVID-19 patients when compared to healthy controls, suggesting a diagnostic role in SARS-CoV2-infected patients [10]. In addition, in multivariate Cox regression analysis, we found a borderline significance between worse survival and PLR (*p* = 0.058) and dNLR (*p* = 0.066). The data on PLR agree with previous findings by Qu R. et al. that reported that the PLR may predict mortality in COVID-19 patients [52]. Interestingly, patients with higher SII values had a significantly worst P/F ratio and chest CT severity score, with no differences regarding the Charlson comorbidity index; this suggests that SII might reflect specifically the pulmonary and respiratory damage occurring in COVID-19 patients rather than a general impairment of their clinical conditions due to comorbidities.

Some limitations of this study should be underlined, including its retrospective nature and the relatively small sample size, which may have impacted the statistical analysis, even if patients were recruited from four different centers, thus improving the generalization of the results. On the other hand, this is the first study to investigate and compare the prognostic roles of a wide range of blood cell count systemic inflammation indexes in COVID-19.

## 4. Materials and Methods

We retrospectively studied 119 COVID-19 patients admitted to the Respiratory Disease and Infectious Disease Units of the University Hospital of Sassari, the Pneumology Unit of the Santissima Trinità Hospital of Cagliari, and the Pneumology Unit of the Mater Olbia Hospital of Olbia, Sardinia, Italy, between 15 March and 15 May 2020. COVID-19 disease was confirmed by reverse transcription polymerase chain reaction (RT-PCR) in all cases. Demographic, clinical, and laboratory data were entered in a dedicated electronic database. In particular, we assessed the following blood cell count inflammation parameters: white blood cell count (WBC), monocytes, lymphocytes, neutrophils, platelets, red blood cell distribution width (RDW), and mean platelet volume (MPV). We then extrapolated combined blood cell indexes of systemic inflammation: the NLR (neutrophil/lymphocyte ratio), dNLR (neutrophils/(white blood cells − neutrophils)), PLR (platelet/lymphocyte ratio), MPR (mean platelet volume/platelet ratio), NLPR (neutrophil/(lymphocyte × platelet ratio)), MLR (monocyte/lymphocyte ratio), SIRI ((neutrophils × monocytes)/lymphocytes), SII ((neutrophils × platelets)/lymphocytes) and AISI ((neutrophils × monocytes × platelets)/lymphocytes). We also collected information regarding the intensity of care received during hospitalization in terms of respiratory support (oxygen supplementation, non-invasive or invasive respiratory support), established parameters of comorbidity (Charlson comorbidity index), hypoxia (PaO_2_/FiO_2_), and lung inflammation severity (chest CT severity score).

The patients were monitored until in-hospital death (non-survivors) or discharge (survivors). The criteria for discharge were (i) fever absence for at least 3 days; (ii) signs of improvement on chest CT scan or X-ray; (iii) two consecutive negative nucleic acid tests performed at least 24 h from each other. The study was conducted in accordance with the declaration of Helsinki and was approved by the ethics committee of the University Hospital (AOU) of Cagliari (PG/2020/10915).

Data are expressed as mean values (mean ± SD) or median values (median and IQR). The Kolmogorov–Smirnov test was performed to evaluate variable distribution. Between-group differences of continuous variables were compared using unpaired Student’s t-test or Mann–Whitney rank sum test, as appropriate. Differences between categorical variables were evaluated by Fisher test or chi-squared test, as appropriate. Receiver operating characteristics (ROC) curve analysis was performed to estimate optimal cut-off values, maximizing sensitivity and specificity according to the Youden index. For survival analysis, time zero was defined as the time of hospital admission.

Survival probability for CBC-derived inflammation indexes was estimated using the means of the Kaplan–Meier curves with the end point being death. Cox proportional hazards regression was performed for both univariate and multivariate analyses. To avoid collinearity bias, the independent prognostic power of the combined blood cell count-derived indexes was separately assessed for each parameter, by correcting for confounders that have a *p* < 0.2 in univariate analysis (age, P/F ratio, intensity of care, and Charlson comorbidity index). Statistical analyses were performed using MedCalc for Windows, version 19.4.1 64 bit (MedCalc Software, Ostend, Belgium).

## 5. Conclusions

The results of this retrospective study showed that the SII is the most significant prognostic biomarker for survival in patients with SARS-CoV2-infected patients when compared to other widely employed blood cell count-derived inflammation indexes such as the NLR, dNLR, PLR, MLR, NLPR, MPR, AISI, and SIRI. Properly designed prospective studies should be performed to confirm the prognostic ability of SII in COVID-19 patients and its utility in the early identification of high-risk patients and the implementation of optimal individualized treatment strategies.

## Figures and Tables

**Figure 1 molecules-25-05725-f001:**
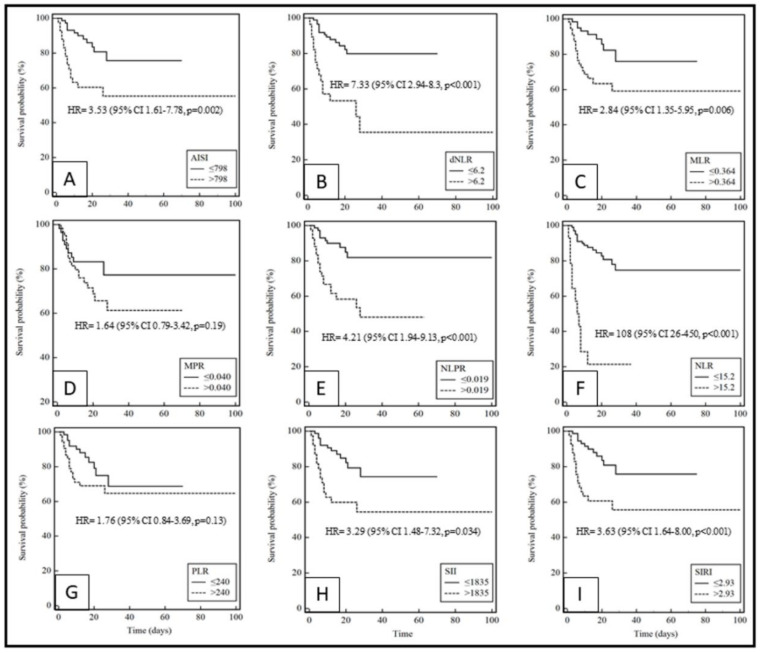
Kaplan–Meier survival curves during hospitalization of COVID-19 patients with different cut-off values of the systemic inflammation indexes investigated. (**A**): AISI; (**B**): dNLR; (**C**): MLR; (**D**): MPR; (**E**): NLPR; (**F**): NLR; (**G**): PLR; (**H**): SII; (**I**): SIRI.

**Table 1 molecules-25-05725-t001:** Demographic, clinical, and hematological features of COVID-19 survivors and non-survivors.

	Global Cohort (*n = 119*)	Survivors (*n = 90*)	Non-Survivors (*n = 29*)	*p*-Value
**Age (Years)**	72 (60–81)	68 (57–78)	80 (74–85)	<0.001
**Gender (F/M)**	42/77	34/56	8/21	0.32
**Smoking Status (No/Yes/Former)**	67/36/16	51/27/12	16/9/4	0.99
**BMI (Non-Obese/Obese)**	92/27	68/22	24/5	0.42
**P/F Ratio**	278 ± 90	301 ± 79	208 ± 88	**<0.001**
**Chest CT Severity Score**	11.0 (6.0–21.0)	10.0 (6.0–19.5)	15.0 (10.5–21.5)	0.12
**Intensity of Care (No, OT, RSni, RSi)**	25/50/22/22	24/37/13/16	1/13/9/6	**0.03**
**Interval between Disease Onset and Admission (Days)**	6.5 (3.0–8.0)	7.0 (4.0–9.0)	4.0 (1.5–6.0)	**0.006**
**Hospital Stay (Days)**	15 (10–26)	20 (13–34)	6 (4–12)	**<0.001**
**Charlson Comorbidity Index**	4.0 (2.0–7.0)	3.5 (2.0–6.0)	6.0 (5.0–8.0)	**<0.001**
**Cardiovascular Disease (No/Yes)**	50/69	44/46	6/23	**0.008**
**Respiratory Disease (No/Yes)**	93/26	72/18	21/8	0.39
**Kidney Disease (No/Yes)**	102/17	77/13	25/4	0.93
**Diabetes (No/Yes)**	94/25	72/18	22/7	0.64
**Cancer (No/Yes)**	101/18	76/14	25/4	0.82
**Autoimmunity (No/Yes)**	113/6	85/5	28/1	0.65
**ACE inhibitors (No/Yes)**	96/23	75/15	21/8	0.20
**ARBs (No/Yes)**	96/23	73/17	23/6	0.83
**WBC (×10^9^ L)**	6.74 (5.00–9.32)	6.47 (5.00–8.66)	9.16 (5.65–13.52)	**0.006**
**Monocytes (×10^9^ L)**	0.35 (0.20–0.50)	0.40 (0.20–0.50)	0.30 (0.26–0.53)	0.85
**Lymphocytes (×10^9^ L)**	0.90 (0.67–1.14)	1.00 (0.70–1.20)	0.70 (0.60–1.03)	**0.012**
**Neutrophils (×10^9^ L)**	5.24 (6.30–7.58)	5.00 (3.56–6.30)	7.30 (3.75–11.15)	**0.012**
**Platelets (×10^9^ L)**	204 (162–265)	211 (166–265)	184 (159–259)	0.43
**RDW (%)**	15.1 (14.1–16.3)	14.9 (13.7–16.4)	15.5 (14.8–16.1)	0.41
**MPV (fL)**	8.45 (8.10–9.00)	8.40 (8.08–8.92)	8.60 (8.18–9.320.37)	0.37

ACE: angiotensin converting enzyme; ARBs: angiotensin II receptor blockers; BMI: body mass index; COVID-19: coronavirus disease 2019; F: female; M: male; MPV, mean platelet volume; OT: oxygen therapy; RDW, red blood cell distribution width; RSi: invasive respiratory support; RSni: non-invasive respiratory support; WBC: white blood cells. All continuous variables are reported as medians and interquartile ranges. Statistical significance set at 0.05. All statistically significant values are reported in bold.

**Table 2 molecules-25-05725-t002:** Blood cell count-derived inflammation indexes of COVID-19 survivors and non-survivors.

	Global Cohort (*n = 119*)	Survivors (*n = 90*)	Non-Survivors (*n = 29*)	*p*-Value
**AISI**	404 (170–1324)	382 (139–894)	952 (201–1943)	**0.025**
**dNLR**	3.45 (2.22–5.77)	3.06 (2.16–5.07)	6.33 (2.87–10.19)	**0.008**
**MLR**	0.364 (0.250–0.566)	0.333 (0.250–0.514)	0.429 (0.297–0.686)	0.058
**MPR**	0.042 (0.036–0.056)	0.040 (0.030–0.056)	0.047 (0.031–0.057)	0.35
**NLPR**	0.0145 (0.0080–0.0249	0.0121 (0.0075–0.0194)	0.0215 (0.0142–0.0508)	**0.0009**
**NLR**	5.67 (3.52–10.37)	5.00 (3.27–8.44)	9.17 (5.35–18.55)	**0.0015**
**PLR**	221 (145–349)	214 (145–339)	265 (144–428)	0.24
**SII**	1137 (645–2393)	1022 (616–1896)	1854 (706–4566)	**0.039**
**SIRI**	1.95 (1.13–4.11)	1.64 (1.07–2.94)	3.11 (1.19–5.56)	**0.010**

AISI, aggregate index of systemic inflammation (neutrophil*platelet* monocyte to lymphocyte ratio); COVID-19: coronavirus disease 2019; dNLR, derived neutrophil to lymphocyte ratio; MLR, monocyte to lymphocyte ratio; MPR, mean platelet volume to platelet ratio; NLR, neutrophil to lymphocyte ratio; NLPR, neutrophil to lymphocyte*platelet ratio; PLR, platelet to lymphocyte ratio; SII, systemic immune-inflammation index (neutrophil*platelet to lymphocyte ratio); SIRI, systemic inflammation response index (neutrophil* monocyte to lymphocyte ratio). All variables are reported as medians and interquartile ranges. Statistical significance set at 0.05. All statistically significant values are reported in bold.

**Table 3 molecules-25-05725-t003:** Receiver operating characteristics (ROC) curves and prognostic accuracy of blood cell count-derived inflammation indexes.

	AUC	95% CI	*p*-Value	Cut-off	Sensitivity (%)	Specificity (%)
**AISI**	0.640	0.546 to 0.726	0.034	>798	59	72
**dNLR**	0.664	0.571 to 0.748	0.014	>6.2	52	85
**MLR**	0.617	0.523 to 0.705	0.054	>0.364	69	57
**MPR**	0.558	0.464 to 0.649	0.37	>0.040	66	52
**NLPR**	0.706	0.615 to 0.787	0.0005	>0.019	66	75
**NLR**	0.697	0.605 to 0.778	0.002	>15.2	38	97
**PLR**	0.572	0.478 to 0.663	0.26	>240	59	58
**SII**	0.628	0.534 to 0.715	0.06	>1835	55	75
**SIRI**	0.659	0.567 to 0.744	0.013	>2.93	59	74

AISI, aggregate index of systemic inflammation (neutrophil*platelet* monocyte to lymphocyte ratio); AUC: area under the curve; CI: confidence interval; COVID-19: coronavirus disease 2019; dNLR, derived neutrophil to lymphocyte ratio; MLR, monocyte to lymphocyte ratio; MPR, mean platelet volume to platelet ratio; NLR, neutrophil to lymphocyte ratio; NLPR, neutrophil to lymphocyte*platelet ratio; PLR, platelet to lymphocyte ratio; SII, systemic immune-inflammation index (neutrophil*platelet to lymphocyte ratio); SIRI, systemic inflammation response index (neutrophil* monocyte to lymphocyte ratio). Statistical significance set at 0.05.

**Table 4 molecules-25-05725-t004:** Hazard ratios of the indexes under investigation obtained by Cox regression analysis.

	HR	95% CI	*p*-Value
Age	1.0134	0.9701 to 1.0588	0.549
P/F Ratio	0.9909	0.9851 to 0.9967	0.002
Intensity of Care	1.0039	0.6568 to 1.5345	0.985
Charlson Comorbidity Index	1.1753	1.0154 to 1.3604	0.030
**AISI**	**1.0000**	**1.0000 to 1.0001**	**0.144**
Age	1.0216	0.9757 to 1.0697	0.362
P/F Ratio	0.9918	0.9857 to 0.9978	0.008
Intensity of Care	0.9429	0.6239 to 1.4249	0.780
Charlson Comorbidity Index	1.1475	0.9862 to 1.3352	0.075
**dNLR**	**1.0926**	**0.9943 to 1.2006**	**0.066**
Age	1.0146	0.9714 to 1.0597	0.515
P/F Ratio	0.9908	0.9850 to 0.9967	0.002
Intensity of Care	1.0044	0.6565 to 1.5365	0.984
Charlson Comorbidity Index	1.1752	1.0172 to 1.3578	0.028
**MLR**	**1.6028**	**0.6277 to 4.0924**	**0.324**
Age	1.0112	0.9691 to 1.0550	0.608
P/F Ratio	0.9903	0.9847 to 0.9959	0.001
Intensity of Care	0.9785	0.6435 to 1.4880	0.919
Charlson Comorbidity Index	1.1798	1.0219 to 1.3621	0.024
**MPR**	**0.3996**	**0.0004 to 363.96**	**0.792**
Age	1.0106	0.9686 to 1.0544	0.627
P/F Ratio	0.9902	0.9846 to 0.9958	0.001
Intensity of Care	0.9698	0.6384 to 1.4732	0.886
Charlson Comorbidity Index	1.1784	1.0200 to 1.3614	0.026
**NLPR**	**0.9669**	**0.0035 to 269.42**	**0.991**
Age	1.0173	0.9727 to 1.0639	0.454
P/F Ratio	0.9914	0.9854 to 0.9974	0.005
Intensity of Care	0.9658	0.6374 to 1.4633	0.870
Charlson Comorbidity Index	1.1582	0.9971 to 1.3454	0.055
**NLR**	**1.0275**	**0.9948 to 1.0612**	**0.100**
Age	1.0191	0.9740 to 1.0663	0.412
P/F Ratio	0.9905	0.9847 to 0.9963	0.001
Intensity of Care	0.9928	0.6474 to 1.5225	0.974
Charlson Comorbidity Index	1.1367	0.9708 to 1.3309	0.111
**PLR**	**1.0006**	**1.0000 to 1.0013**	**0.058**
Age	1.0213	0.9754 to 1.0694	0.369
P/F Ratio	0.9913	0.9854 to 0.9972	0.004
Intensity of Care	1.0069	0.6553 to 1.5469	0.975
Charlson Comorbidity Index	1.1299	0.9651 to 1.3229	0.129
**SII**	**1.0001**	**1.0000 to 1.0001**	**0.029**
Age	1.0146	0.9710 to 1.0601	0.517
P/F Ratio	0.9912	0.9853 to 0.9971	0.004
Intensity of Care	1.0127	0.6621 to 1.5489	0.954
Charlson Comorbidity Index	1.1767	1.0166 to 1.3621	0.029
**SIRI**	**1.0279**	**0.9885 to 1.0688**	**0.167**

AISI, aggregate index of systemic inflammation; CI: confidence interval; COVID-19: coronavirus disease 2019; dNLR, derived neutrophil to lymphocyte ratio; HR: hazard ratio; MLR, monocyte to lymphocyte ratio; MPR, mean platelet volume to platelet ratio; NLR, neutrophil to lymphocyte ratio; NLPR, neutrophil to lymphocyte x platelet ratio; P/F: PaO2/FiO2; PLR, platelet to lymphocyte ratio; SII, systemic immune-inflammation index; SIRI, systemic inflammation response index. All statistically significant values are reported in bold.

**Table 5 molecules-25-05725-t005:** Demographic, clinical, and laboratory characteristics of COVID-19 patients stratified by the SII.

	SII ≤1835 (*n = 81*)	SII >1835 (*n = 38*)	*p*-Value
**Age (Years)**	71 (58–80)	74 (62–80)	0.57
**Gender (F/M)**	32/49	10/28	0.16
**Smoking status (No/Yes/Former)**	48/25/8	19/11/8	0.24
**BMI (Non-Obese/Obese)**	62/19	30/8	0.77
**P/F Ratio**	301 ± 85	226 ± 79	**<0.001**
**Chest CT Severity Score**	9.0 (5.0–14.0)	21.0 (13.3–23.0)	**<0.001**
**Intensity of Care (No, OT, RSni, RSi)**	16/39/12/14	9/11/10/8	0.21
**Interval between Disease Onset and Admission (Days)**	6.0 (3.0–8.0)	6.0 (3.0–8.8)	0.99
**Hospital Stay (Days)**	17 (11–26)	14 (6–27)	0.29
**Survivors (No/Yes)**	13/68	16/22	**0.002**
**Charlson Comorbidity Index**	4.0 (2.0–6.3)	4.5 (2.0–7.0)	0.23
**Cardiovascular Disease (No/Yes)**	35/46	15/23	0.7
**Respiratory Disease (No/Yes)**	67/14	26/12	0.08
**Kidney Disease (No/Yes)**	69/12	33/5	0.81
**Diabetes (No/Yes)**	62/19	32/6	0.34
**Cancer (No/Yes)**	70/11	31/7	0.49
**Autoimmunity (No/Yes)**	77/4	36/2	0.94
**ACE inhibitors (No/Yes)**	64/17	32/6	0.51
**ARBs (No/Yes)**	65/16	31/7	0.86
**WBC (×10^9^ L)**	5.90 (4.81–7.87)	9.90 (7.49–12.60)	**<0.001**
**Monocytes (×10^9^ L)**	0.39 (0.20–0.50)	0.30 (0.28–0.50)	0.77
**Lymphocytes** **(×10^9^ L)**	1.00 (0.80–1.35)	0.64 (0.50–0.80)	**<0.001**
**Neutrophils** **(×10^9^ L)**	4.40 (3.40–5.55)	8.65 (6.00–11.60)	**<0.001**
**Platelets (×10^9^ L)**	183 (147–233)	267 (220–361)	**<0.001**
**RDW (%)**	15.4 (14.2–16.6)	14.9 (13.7–16.0)	0.14
**MPV (fL)**	8.45 (8.20–9.00)	8.45 (7.9–8.9)	0.38

ACE: angiotensin converting enzyme; ARBs: angiotensin II receptor blockers; BMI: body mass index; COVID-19: coronavirus disease 2019; F: female; M: male; MPV, mean platelet volume; OT: oxygen therapy; P/F: PaO2/FiO2; RDW, red blood cell distribution width; RSi: invasive respiratory support; RSni: non-invasive respiratory support; WBC: white blood cells. All continuous variables are reported as medians and interquartile ranges. Statistical significance set at 0.05. All statistically significant values are reported in bold.

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
