# Peer review of "The Systemic Inflammation Index on Admission Predicts In-Hospital Mortality in COVID-19 Patients"

_molecules, 2020, doi:10.3390/molecules25235725_

Round 1
Reviewer 1 Report
This is a nice study, well-designed and well-written. Although the authors have not used some of the well-known poor prognostic factors for COVID-19, i.e. D-dimers, IL-6 or CRP, I think that the use of every-day practice markers, very easy to be obtained, is extremely useful.
However, I would advise the authors to discuss and cite other important prognostic scores including that of Gerotziafas et al. Thromb Haemost. 2020 Sep 22. doi: 10.1055/s-0040-1716544.
Other papers that have to be cited and discussed are: Intensive Care Med. 2020 Oct 29:1-14; Am J Hematol. 2020 Jul;95(7):834-847; Clin Exp Med. 2020 Nov;20(4):493-506.
Author Response
We would like to thank the reviewer for the valuable suggestions. We added and cited the articles suggested, expanding the reference list of the manuscript and, thus, allowing the readers to find more papers on the topic. The focus of our paper was on blood cell count derived indexes, therefore we did not discuss papers proposing different composite indexes. The parts and citations added in the manuscript are highlighted in yellow.
Reviewer 2 Report
The study evaluated factors predictive of mortality in hospitalized patients with COVID 19 pneumonia, searching them within inflammation status derived by blood cell reactivity to the infection, looking at parameters not properly tested before in the authors’ opinion.
They found the SII only (systemic immune-inflammation index (neutrophil*platelet to lymphocyte ratio)), after correction for clinical presentation of the disease, with a p= 0.03 is predictive of death of 29 patient, with a retrospective approach.
Many other reports investigated this topic, in a similar manner (please see: Rokni M, Ahmadikia K, Asghari S, Mashaei S, Hassanali F. Comparison of clinical, para-clinical and laboratory findings in survived and deceased patients with COVID-19: diagnostic role of inflammatory indications in determining the severity of illness. BMC Infect Dis. 2020 Nov 23;20(1):869, and the review: Tjendra Y, Al Mana AF, Espejo AP, Akgun Y, Millan NC, Gomez-Fernandez C, Cray C. Predicting Disease Severity and Outcome in COVID-19 Patients: A Review of Multiple Biomarkers. Arch Pathol Lab Med. 2020 Aug 20).
Minor: the readability of Table 5 is poor, please reformat conveniently; please indicate, in the abstract and in the conclusion, the retrospective design of the study; please discuss the relevance of known indexes of clinical presentation and respiratory involvement (Charlson index and P/F) with respect to SII.
Author Response
We would like to thank the reviewer for the valuable suggestions. We added and cited the articles suggested.
Minor issues.
Table 5 summarizes the same parameters as Table 1 in patients with higher and lower SII cut-off values; we preferred to leave it unchanged in order to avoid confusion due to heterogeneity with Table 1. The retrospective design of the study was added in the abstract and conclusion of the manuscript. We also added a brief comment on the relationship between clinical indexes and SII in COVID-19. The parts and citations added in the manuscript are highlighted in yellow.